# Mobilizing Benefit-Sharing Through Transportation Infrastructure: Informal Roads, Extractive Industries and Benefit-Sharing in the Irkutsk Oil and Gas Region, Russia

**Vera Kuklina [1,2,*], Andrey N Petrov [3,4], Natalia Krasnoshtanova [5] and Viktor Bogdanov [6]**

1   Department of Geography, George Washington University, Washington, DC 20036, USA
2   Lab of Natural Resource Management and Political Geography, V.B. Sochava Institute of Geography SB RAS, 664033 Irkutsk, Russia
3   ARCTICenter and Department of Geography, University of Northern Iowa, Cedar Falls, IO 50614-0406, USA; andrey.petrov@uni.edu
4   Department of Social Sciences and Humanities, Russian State Hydrometeorological University, 192007 Saint Petersburg, Russia
5   Lab of Theoretical Geography, V.B. Sochava Institute of Geography SB RAS, Irkutsk 664033, Russia; knesun@mail.ru
6   Lab of Cartography, Geoinformatics and Remote Sensing, V.B. Sochava Institute of Geography SB RAS, 664033 Irkutsk, Russia; victvss@gmail.com
*   Correspondence: kuklina@gwu.edu

**Abstract:** Road infrastructure development is an existing, but not a frequent element of extractive industry benefit-sharing frameworks in remote northern regions. However, it is often at the center of extractive activity and inflicts major impact on environment and communities. This paper examines the benefits and impacts derived from development of informal roads, i.e., vehicular roadways beyond the current publicly-governed road networks constructed, maintained and/or used by various entities and individuals based on private, special purpose and/or informal practices and regulations. Based on several field studies, GIS analysis of road networks and examination of secondary sources, the article investigates the use of informal roads as a form of benefit-sharing and details their impact on mobilities, environment and livelihoods of local and indigenous communities in the Irkutsk Oil and Gas region, Russia. We argue that construction, maintenance and use of the industry-built roads can be a part of benefit-sharing agreements, albeit mostly semi-formal and negotiated. The gains and problems stemming from 'trickle-down' (i.e., unintended) effects of the road networks are the most significant. The community-relevant implications of informal roads go far beyond immediate impacts on surrounding environment, but deeply affect subsistence activities, mobility, food security, personal safety and even consumer preferences of the indigenous residents.

**Keywords:** informal roads; benefit-sharing; extractive industries; transportation infrastructure; indigenous people

## 1. Introduction

Benefit-sharing can be defined as the distribution of monetary and non-monetary benefits generated through resource-extraction activities [1,2]. Benefit-sharing is closely related to the notions of Corporate Social Responsibility (CSR) and Social License to Operate (SLO) and recognizes the right of local communities to receive a share of profits received by a resource company [2]. The concept

highlights the necessity to share these benefits with the local stake- and rightsholders who live near to the resource extraction areas and provide access to the resource for companies.

Benefit-sharing aims to improve the local communities' wellbeing and promotes fate-control and self-reliance [3]. Benefit-sharing can be implemented by the state by means of sovereign wealth funds, land lease agreements, social investments or by extractive companies' own initiatives, such as philanthropic, (supporting cultural festivities, revitalization of languages or sports), investments in infrastructure, education, training and technology, as well as in new jobs creation [2,4].

Benefit-sharing regimes that characterize the relationships between extractive companies and local communities rely on various mechanisms of benefit implementation. These mechanisms are financial, legal and procedural ways used to operationalize benefit-sharing [2]. In the literature, four primary mechanisms prevalent in the Arctic have been identified: streamlined (mandated), negotiated, semi-formal and "trickle-down".

Streamlined or mandated benefits are firmly established and institutionalized in legislative or other legal acts and prescribe obligations, responsibilities, specific implementation tools and often give access to remedy, if conditions are not met. Many of these benefits are legally-binding, such as taxes and royalties.

Negotiated benefits refer to negotiated arrangements between companies and/or governments and/or communities. These negotiations lead to formalized agreements that have various levels of legal standing and nature. Classic examples of negotiated benefit-sharing arrangement is an Impact and Benefit Agreements (IBAs) [5,6] and socio-economic cooperation agreements [7].

Semi-formal benefits are relatively informally established forms of compensation, investment and sponsorships often originated upon specific request and, in Russia, manifested in the form of the "plea-and-take" system [8]. An extractive company typically receives such requests from communities or government entities and decides whether it chooses to share benefits. This process is intransparent and decision-making power is always retained by a company.

In the article, we offer a look at how benefit-sharing mechanisms are materialized in a specific context, in this case, expansion of road networks. Road infrastructure development is an existing, but not frequent element of benefit-sharing arrangements. It is typically overshadowed by other provisions, such as compensation payments, grants, investment in social infrastructure, etc. However, roads are the main element of transportation infrastructure in remote regions, and as such, are often at the center of extractive activity and inflict a major impact on the environment and communities [9,10]. As a result, both construction and use of the industry-built roads can be a part of benefit-sharing agreements, albeit mostly semi-formal and negotiated. The gains (and problems) stemming from "trickle-down" (i.e., unintended) effects of the road networks are, perhaps, by far the most significant. Since many extractive activities require an extensive and expensive transportation network development, it is natural to expect that roads will be playing a key role as the elements of benefit-sharing frameworks. While extractive companies facilitate local mobilities via the provision of transport services, road maintenance, or oil and gas supplies, other aspects of increasing human and non-human mobilities remain beyond the scope of negotiations on benefit-sharing.

Transportation infrastructure, and roads specifically, are often considered important factors of economic development in remote areas [11,12], although they can also play a dual role by generating undesirable consequences [13]. They cross multiple domains of human use and well-being: not just mobility and access, but also food security, personal safety, human/Indigenous rights, health, and knowledge. By improving access and increasing mobility new roads reduce the cost of travel, open new opportunities to reach markets and exploit local resources, provide connectivity among communities, and enhance the delivery of transport-dependent services. In this respect, road infrastructure development is often viewed as a significant benefit to local communities [14,15]. At the same time, road infrastructure can exert negative impacts on local communities and the environment. Some forms of traditional mobilities are disrupted and hindered by new infrastructural development which is sometimes conceptualized as "infrastructural violence" [13,16]. There are

numerous examples illustrating roads as undesirable for local communities and harmful for the environment [17]. Remoteness and extreme climate conditions lead to use of non-local workforces in extractive development areas and result in changes in mobilities [18–21]. The negative effects of transportation infrastructure on traditional cultures and lifestyles arise from increasing access by "outsiders" to the local resources (e.g., traditional hunting and gathering grounds), exposing local communities to social problems (e.g., alcoholism, drug use, violence etc.) and changes in family and community life, subsistence economy and traditional cultures in general [22–26]. In shamanic worldviews, characteristic for many indigenous groups, people upturning the ground are expecting misfortunes. Road construction in this perspective violates spirits' dignity. Infrastructural elements within this perspective of responsive landscapes are articulated as "scars" [27].

The studies focusing on environmental changes identify transportation infrastructure as one of the main contributors of surface disturbance and habitat fragmentation that impair access of local communities to subsistence resources (e.g., [9,28,29]). The roads are often linked to land-use change and fragmentation, deforestation, pollution and threat to biodiversity. Permanent roads negatively affect the hydrologic regime, permafrost, vegetation, and contribute to pollution (e.g., [23,29–34]). Researchers call for preserving the roadless areas to prevent "contagious development" that the road construction brings [35].

In the Arctic, infrastructural development has been the main state endeavor that first brought large flows of population to the remote regions during the construction stage, and then facilitated the development of extractive industries in the areas formerly too remote for exploration. Depending on the distance from the cities, researchers find communities having more or less access to basic services and goods [36–38]. Alternative modes of transportation available through new infrastructure development also reduce the dependence on specific modes of transportation and thus diversify economic activities [39]. Currently, not only state, but also other actors, such as extractive companies and local communities are engaged in either developing or utilizing roads in formal or informal ways. For example, anthropologists document the road networks formed by indigenous people different from the ones formed by the states [40,41]. These roads usually neither exist officially, nor mapped [42]. In Siberia, we find the importance of informal roads not only for indigenous people, but also for other remote communities lacking political power.

Informal roads are vehicular roadways beyond the current publicly-governed road network constructed, maintained and/or used by various entities and individuals based on private, special purpose and/or informal practices and regulations. In the literature, they are documented as "hidden roads" [43], "unofficial roads" [44–46], "unchartered roads" [47] or "roads of local significance" [48]. To distinguish informal roads from planned and constructed formal routes, Trombold [49] proposed the term "informal roads". However, the difference between formal and informal roads is not obvious: once constructed formal roads may become informal if left unmaintained due to changes in priorities and budgets. For example, Argounova-Law [50] emphasized social significance as the main feature defining roads. Moreover, the proliferation of modern all-terrain and 4WD motorized vehicles in remote parts of the world imposes less stringent requirements on road construction and maintenance [51], while affecting the environment [52–54].

With increasing technological development and connectivity, mobilities are afforded not only by conventional official roads, but also other linear structures created for initial exploration, construction and maintenance of the infrastructural objects that remain usable after the initial purpose was abandoned. Together with other private and special purpose roads they form a network of transportation infrastructure that is not existent on official maps nor it is governed by official documents and authorities. Depending on their use, users and impact on local communities and environment, these roads can be differently represented in benefit-sharing.

The focus on informal roads allows to explore how benefit-sharing is materialized in specific landscapes, and how they are distributed and experienced by members of local communities. Specifically, we pursue the answers to the following questions: (1) what is the role and nature

of informal roads as part of benefit-sharing? and (2) how different benefit-sharing mechanisms mobilize benefits stemming from the informal roads infrastructure, and what issues do they create? To accomplish this, we examine the uses of informal roads as a form of benefit-sharing and investigate how they affect mobilities, environment and livelihoods of local and indigenous communities in the Irkutsk Oil and Gas region.

## 2. Materials and Methods

For identification of informal roads, we used existing Russian federal regulations. The norms and rules of the automobile road construction and exploitation are regulated by the Federal Law #257 [55]. The public roads according to their significance and jurisdiction are classified into federal, regional and inter-municipal, municipal, and private roads [56]. Among public roads only ones connecting regional centers have federal significance, and therefore receive federal funding and, consequently, are better maintained. However, even some federal roads lack maintenance, such as part of the seasonal winter road "Viliuy" which connects the Republic of Sakha (Yakutia) with Ust-Kut and a permanent federal road network [57].

In terms of access, the federal law distinguishes roads with public (unlimited) and non-public access. The distinction between these two kinds of roads is defined only by the availability of specific equipment restricting access. If the road is not fenced and monitored, it is available for public access. For example, the forest roads are regulated by specific rules of design and construction [58]. According to these regulations, forest roads, as well as other service roads, are not designated for public use and general vehicles. However, absence of fences or gates allows people to use them almost without restrictions. When private forest companies rent specific forest areas, they often construct checkpoints. Since access to some private roads and state-owned non-public roads can be negotiated and varies from case to case, we consider these roads as informal. Finally, informal roads consist of former public or private roads that were abandoned, but continue to be used for travel; geophysical line clearings—roads made once for geological exploration and then abandoned or used for other purposes; trails and tracks traditionally used for subsistence activities and used by motorized vehicles; and unofficial tracks or roads of various qualities laid between settlements that have not been recognized, and therefore, maintained by authorities.

In order to capture the diversity of informal roads, their users and types of use within benefit-sharing arrangements, we combined interviews, participant observations and GIS analysis of road networks. The field studies were conducted in 2014 and 2016 with a focus on interactions between local communities and extractive industry, and in 2019 with a focus on informal roads and their users. The interviewees were found using snowball method and former social networks. The duration of interviews ranged from twenty to ninety minutes and averaged forty to sixty minutes. Although during these years we gathered 55 interviews in the study area, for analysis in this paper we included 16 in-depth interviews with local hunters, community leaders, municipal authorities and company representatives (Table 1), as well as materials of participant observations in the villages of Vershina Khandy (Kazachinsko-Lenskii Rayon), Tokma (Katangskii Rayon) and city of Ust-Kut (Ust-Kutskii Rayon) (the research is exempt from IRB review under DHHS regulatory category 2 (IRB# NCR191103)). In particular, we analyzed the interviews where respondents discussed the benefits and problems of transportation accessibility, environmental degradation, and mobility brought by development of the networks of automobile roads. The interviews were transcribed, anonymized and coded using NVivo to explore specific discourses related to the specific roads, their users and uses. The research is in part based on grounded theory as it is based on an analysis of previous interviews for formulation of the following research objectives. On the other hand, the research is based on synthesis of existing studies of benefit-sharing mechanisms. For more empirical grounding we also used public environmental impact assessment materials for Kovyktinskoie gas deposit [59], and annual socio-economic reports produced by municipalities.

**Table 1.** Key study sites and interviewees.

| Community | Number of Interviewees | Local Hunters (LH) | Community Leaders (CL) | Company Employees (CE) | Government Officials (GO) |
|---|---|---|---|---|---|
| Vershina Khandy | 3 | 2 | 1 | - | - |
| Tokma | 8 | 4 | 2 | - | 2 |
| Ust-Kut | 5 | - | - | 2 | 3 |

There is some disbalance in the representation of different groups across local communities related to the structure, size and stage of industrial exploration of communities. In particular, the village Vershina Khandy does not have any representatives of government because officially it is considered as an inter-settlement territory with an unstable population which varies from five to a few dozens, according to local experts. We did not have interviews with representatives of companies operating in the areas of Tokma and Vershina Khandy, because they are located beyond the study area. In case of Ust-Kut, hunting plays a marginal role in the city, and there are no formally organized communities beyond the ones related to the government.

*Study Area*

The case study area (Figure 1) remains one of the remote places lacking transportation infrastructure. As in many other Arctic and Subarctic communities, accessibility often has seasonal character: some remote communities in summer use water routes and in winter—winter roads. The presence of permafrost over most of the area increases the cost of road construction and maintenance by 3 to 5 times compared to temperate regions [60]. The study area is dominated by the typical Siberian boreal forest with boggy landscapes that makes moving around difficult even for off-road vehicles and snowmobiles. High costs of construction and maintenance make authorities to look for different ways to reduce costs. The local public automobile roads have only regional significance which means the intensity of traffic is between 200–2000 vehicles a day [56]. The regions with seasonal transportation access receive subsidies for transportation of food and fuel which is cheaper than building new roads. Therefore, companies, interested in development in those areas have to build their own roads to move people and goods.

Our study sites, the villages Vershina Khandy and Tokma, are settled by Evenki and Russian old-settlers. Evenki, indigenous Tungus-speaking people traditionally conducted reindeer herding, hunting, fishing, and gathering Siberian pine nuts, berries and herbs. In the 18th century Cossacks arrived in the area. Among the settlements formed during that time was the city of Ust-Kut (44,500 residents) founded in 1631 on the Lena River by migrants from European Russia. The city of Ust-Kut has developed as a key transport hub with the construction of the Osetrovskiy River Port (the largest in the USSR) in 1950 and the Baikal–Amur Mainline (BAM) railway in 1975. The main sources of income local population derive from the public sector, oil industry, transportation, forest industry, and service sector [8].

The villages of Tokma and Vershina Khandy were founded in the 19th century by Russian settlers and evolved in the early 20th century as Russian trade posts (factories), supplying hunters with food, weapons, and other goods, as well as buying products of hunting activity from hunters. Gradually, the local nomadic population of Evenki settled near these posts. In the 1930s, when collectivization began, the sedentarization process accelerated. The first hunting organizations were formed here at the same time. During the Soviet period, commercial hunting in the northern districts of the Irkutsk Region was controlled by the state enterprises [61,62]. However, with the collapse of the Soviet Union, state hunting enterprises were closed, and indigenous people faced the need to independently protect their rights to traditional lands and activities. To preserve traditions of Evenki land use in the 1990s and the beginning of the 2000s, the *obshchinas*, indigenous non-governmental enterprises, were organized and

obtained hunting licenses on the territories of their traditional land use. Nowadays commercial hunting remains the source of cash income for many Evenk and old-settler hunters of these villages [63].

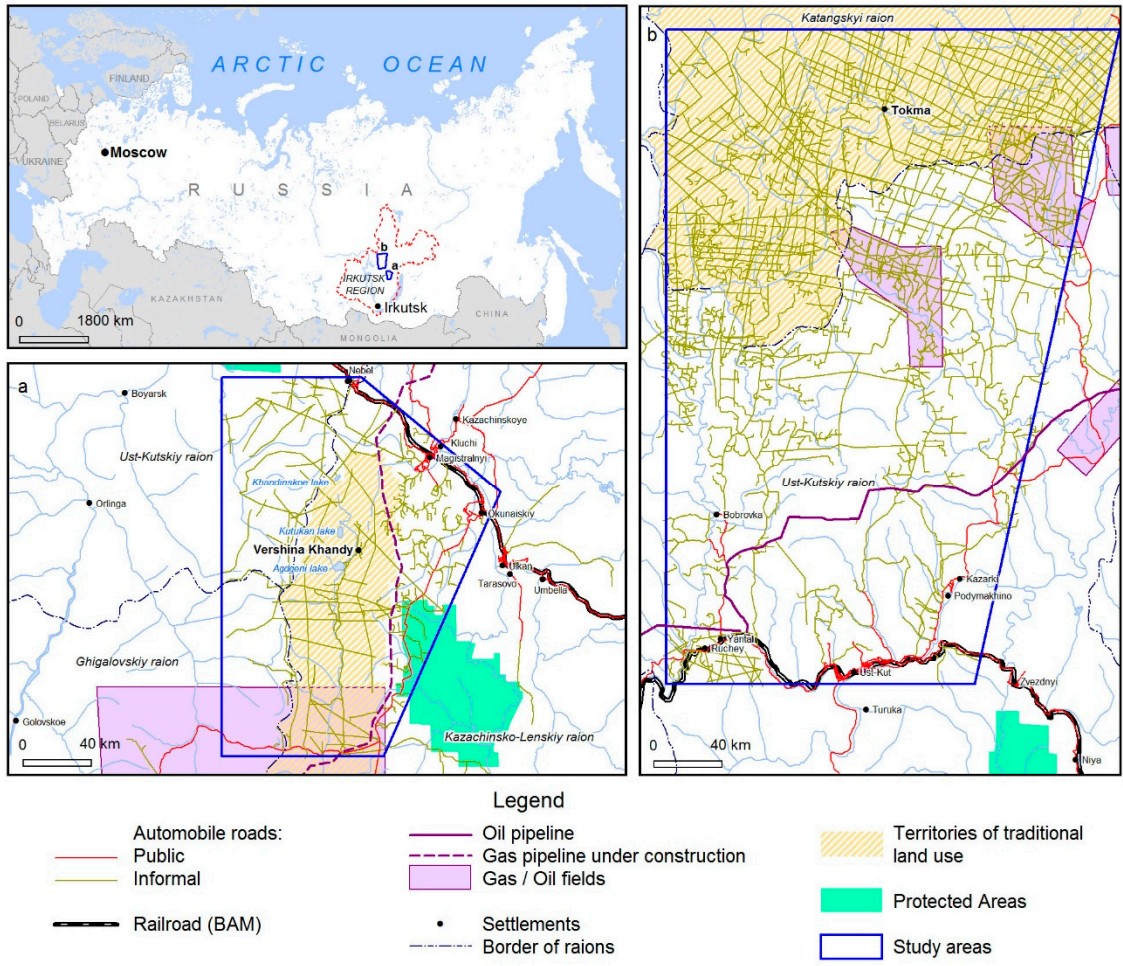

**Figure 1.** Study area.

The first informal road networks were formed along straight lines (forest clearings, so-called 'profiles') made in the taiga for geophysical exploration in the 1970s. They were successfully included in the hunting infrastructure by local hunters for driving on snowmobile. In these years first winter car road connected Tokma with other settlements and cities. In the mid-2000s, the informal road network was expanded with oil and gas exploration, the forest logging industry and supporting service roads (Figure 1).

The main extractive companies in the area near Tokma are Irkutsk Oil Company (IOC) which operates on the Yaraktinskoie and Ichodinskoie oil deposits, and "Russian Forest Group" (former TSLK). Vershina Khandy traditional territories are used for gas exploration sites and gas pipeline construction by Gasprom, as well as for forestry by the state-owned Forest Service, Russian companies Rusforest and Kirenskles, and a Chinese-owned company Eurasia. The Kovyktinskoie gas condensate deposit in the traditional territory of Vershina Khandy Evenki was discovered in 1987 and is known as the biggest gas deposit in the Russian East [64]. In 2014, an agreement between Gasprom and the Chinese National Petroleum Company (CNPC) on the construction of the Power of Siberia gas pipeline from the deposit was signed at the highest level [65]. It is planned to be the main source of gas transported to China in the future. After the gas pipeline will be built to Kovyktinskoie gas condensate deposit (by 2023), China will become the second-largest importer of Russian gas.

## 3. Results

### *3.1. Benefit-Sharing Arrangements and Informal Roads*

In the study area, informal roads become a part of benefit-sharing arrangements that employ different mechanisms. Since these transportation networks lack formalized regimes of use and maintenance, the mechanisms by which they can be engaged in benefit-sharing frameworks will likely not be streamlined or mandated, but rather negotiated, semi-formal and "trickle-down". In other words, informal roads, either directly or not, could be a part of a benefit-sharing "package" that is more or less formally negotiated between a company and community or unintentionally emerged as a result of infrastructure development. Table 2 shows different mechanisms and provides examples of benefits and accompanying issues associated with each form of benefit-sharing in the Irkutsk Oil and Gas region. Detailed case studies are also discussed to illustrate the role of informal roads in benefit-sharing arrangements based on data from Tokma, Vershina Khandy and Ust-Kut.

**Table 2.** Mechanisms of benefit-sharing in the Irkutsk Oil and Gas region.

| Mechanisms | Benefits | Issues | Benefit Sharing Implications |
|---|---|---|---|
| Streamlined/Mandated | not applicable | | |
| Negotiated | Access to company-administered roads, negotiations for local use, part of SE partnerships for investment in informal roads (maintaining or cleaning community roadways) Compensation for disturbance and damage associated with road construction and operation | Relinquishment of transit rights over land (full or partial) Acceptance of disturbance and damage Deceptive negotiation practices Lack of negotiating capacity Lack of access to enforcement and remedy | Can be considered for compensation payments, special access rights and privileges as a part of benefit-sharing arrangements |
| Semi-formal | Ad hoc access and use, sponsorship of maintenance beyond SEPA Tolerance to undetected or illicit use by the locals | Lack of control and high uncertainty Propagation of dependency Danger of fines and prosecution Subject to surveillance and violation of privacy Restriction of mobility and securitization | Formalizing access and use rights as a component of benefit-sharing |
| Trickle-down/derived | Time, fuel savings, increased accessibility and mobility, recreational and tourist access | Dependency on company's will to have roads open. Uncertainty of use. Lack of purposeful benefit-sharing (e.g., road ends in a few km from a village) | Predicting and monitoring these effects as a part of benefit-sharing frameworks |

### 3.1.1. Negotiated Benefits

Negotiated benefits result from negotiations between extractive companies, government authorities and local communities. In cases when transportation infrastructure is a part of benefits covered under such agreements, it typically concerns the access to company-administered roads, availability for local use, or investment in informal roads construction or maintenance. On the other hand, roads developed by extractive companies may trigger a negotiated compensation for disturbance and damage.

For example, near Tokma and Ust-Kut, the newly built oil drilling infrastructure is connected with the "Eastern Siberia–Pacific Ocean" (ESPO) pipeline. It consists of the oil pipelines and service roads which usually have restricted access. The service road constructed along the ESPO pipeline after negotiations with local and regional authorities of the Republic of Sakha (Yakutia), where this road continues, became the main transportation pathway for vehicles delivering goods to the region. The local residents in the Irkutsk region and in Sakha are allowed to use the road for free if they can prove their local residence, however, the cargo transportation is charged additional fees.

Near Vershina Khandy, a Gazprom subsidiary is building 14 bridges and constructing 80 kilometers of the gravel road to connect the Kovyktinskoie gas condensate deposit and its shift-worker camp [66]. The road leads from the BAM railway to the Gasprom shift-worker camp while the remaining part allows movement of vehicles with the speed of around 20 km/h due to bumps and potholes in summer.

In winter, when bumps and potholes covered with snow, the traffic increases. It is the shortest road from Irkutsk to such cities as Severobaikalsk (population 22,000) and Kirensk (population 11,000) and is used by public transport.

As for the forest companies, the Evenki obshchina in Vershina Khandy receives compensation only from RusForest (16,000 Rubles or about $500 USD) plus wood supply annually [67]), a forestry company that also hires residents to work at their checkpoint in summer. In Tokma, in 2010 another forest operator, TSLK, concluded an agreement with the local obshchina where, beyond other benefits, the company took responsibility to maintain a winter road connecting the village with the TSLK forest road [68].

In addition, agreements between companies and local communities often have other components related to the road network development. In particular, the Tokma obshchina and IOC have signed an agreement according to which the company guarantees to supply gas condensate to the local hunters and to transfer money for hunting licenses [68]. However, the hunters should manage to transport the fuel from the company's Yaraktinskoie oil field which is located more than 100 kilometers by the company's winter service roads. They usually hire a tank truck to deliver fuel that costs around 1000 Rubles for 200 liters [69]. It is cheaper than diesel fuel which costs around 10,000 Rubles for 200 liters in the northern districts. However, the gas condensate does not have stable quality and sometimes hunters are hesitant to use it due to the risk of damage to their vehicles.

The purchase of transportation vehicles (on- and off-road) is the preferred way to receive compensation and benefits from oil and gas companies by the Vershina Khandy obshchina members. They received first used off-road vehicle in the early 2000 s from the Rusia Petroleum. In 2017, they received 5.5 million Rubles in compensation from Gasprom to cover administrative expenses for the functioning of obshchina and maintenance of their territory of traditional land use and buy transportation vehicles, such as snowmobiles and swamp buggies [67].

### 3.1.2. Semi-formal Benefits

Semi-formal benefit-sharing includes interactions between extractive companies' representatives with members of local communities related to construction, use and maintenance of the informal road. For instance, to get to the village, Vershina Khandy residents use parts of the forest road currently rented by RusForest. The road is closed for public in summer, but based on a tacit agreement with the company, the residents of the village are allowed to enter. However, the forest road ends in about 15 kilometers from the village. During the BAM construction, the locals asked a bulldozer driver to beat a track to the village through the forest which he did, but in about 8 km to the village, the boggy area begins, which bulldozer could not overcome.

In Vershina Khandy, the service roads to the gas deposit are used by hunters while officially they are closed for public access:

"There is Kovyktinskaia road to the deposit. And our hunters are there. We agreed [with the company] they would have access to their hunting grounds. Once N. complained that [company] restricted access. I called the principal project engineer and asked why they closed access to hunters. He said they didn't. Hunters just need to confirm they are from our obshchina and they will be given access" [67].

Frequent encounters between extractive industry workers and local hunters develop personal relations which can be of mutual benefit. For instance, a worker can informally negotiate with a hunter exchange of fuel for some other goods (traditional products or services). As in other remote regions (e.g., [50]), company drivers have given assistance to anyone who got stuck or other troubles on those roads. Since the main users of these road networks are extractive companies' workers and hunters, interdependence grows in this direction.

Since the winter roads are highly dependent on the weather conditions, scheduled road maintenance according to agreements may not be enough for local mobility. Cleaning of informal roads is another area where semi-formal arrangements play a prominent role. In Tokma, Irkutsk Oil Company occasionally clear snow on the winter road based on community requests: "Well, they usually do not mind clearing the winter road. However, we need to make requests" [68].

Since the roads are made informally, they do not have other elements of road infrastructure, such as road signs etc. As a result, it is easy to get lost in their labyrinths (see Figure 1). Usually, hunters privately negotiate with the companies working on their hunting plots to receive maps of informal roads and immediate plans for their development, to plan hunting activities [70].

### 3.1.3. "Trickle-down" Benefits

"Trickle-down" benefits are typically associated with unplanned positive effects of the roads. In the study region, extractive companies have invested in improving existing roads, turned parts of former seasonal roads into permanent ones, built new roadways and made tracks for vehicles in previously inaccessible areas. All these endeavors significantly improve local accessibility and mobility and open prospects for the development of road services. Areas, previously heavily dependent on seasonal roads and much more expensive air transportation, have secured delivery of goods and services for comparatively lower prices. The reduction of travel time and distance has been also significant that allowed to save on fuel, vehicle maintenance, transportation costs and other related expenses.

One example of increased transportation accessibility is access to the ESPO service road. It is estimated that travel time decreased several times for the trips from Ust-Kut to the Republic of Sakha (Yakutia). As a result, local and non-local drivers, as well as transport companies, found new ways of offering transportation services. Some residents have opened road cafes and repair stations along these major informal roads. Figure 2 demonstrates a drastic improvement in transportation accessibility in remote areas north of Ust-Kut after service roads were built. As a result, the city increased its importance as a key area in transporting goods to Yakutia and servicing major extractive industry developments in Eastern Siberia. Due to the new construction of all-season and especially winter roads, places previously accessible only within 30–40 hours of driving are now reachable in half of that time.

At the local level, informal roads are used by local hunters in their traditional activities to move around or to organize hunting. As one hunter noted, the geophysical line clearings are better for travel on snowmobiles than his own trails: he sets up traps along the roads and finds that some animals also prefer to move there [71]. Some residents of neighboring settlements and even distant cities have benefitted from improved access to recreational hunting and fishing activities as well as to other tourist purposes [72].

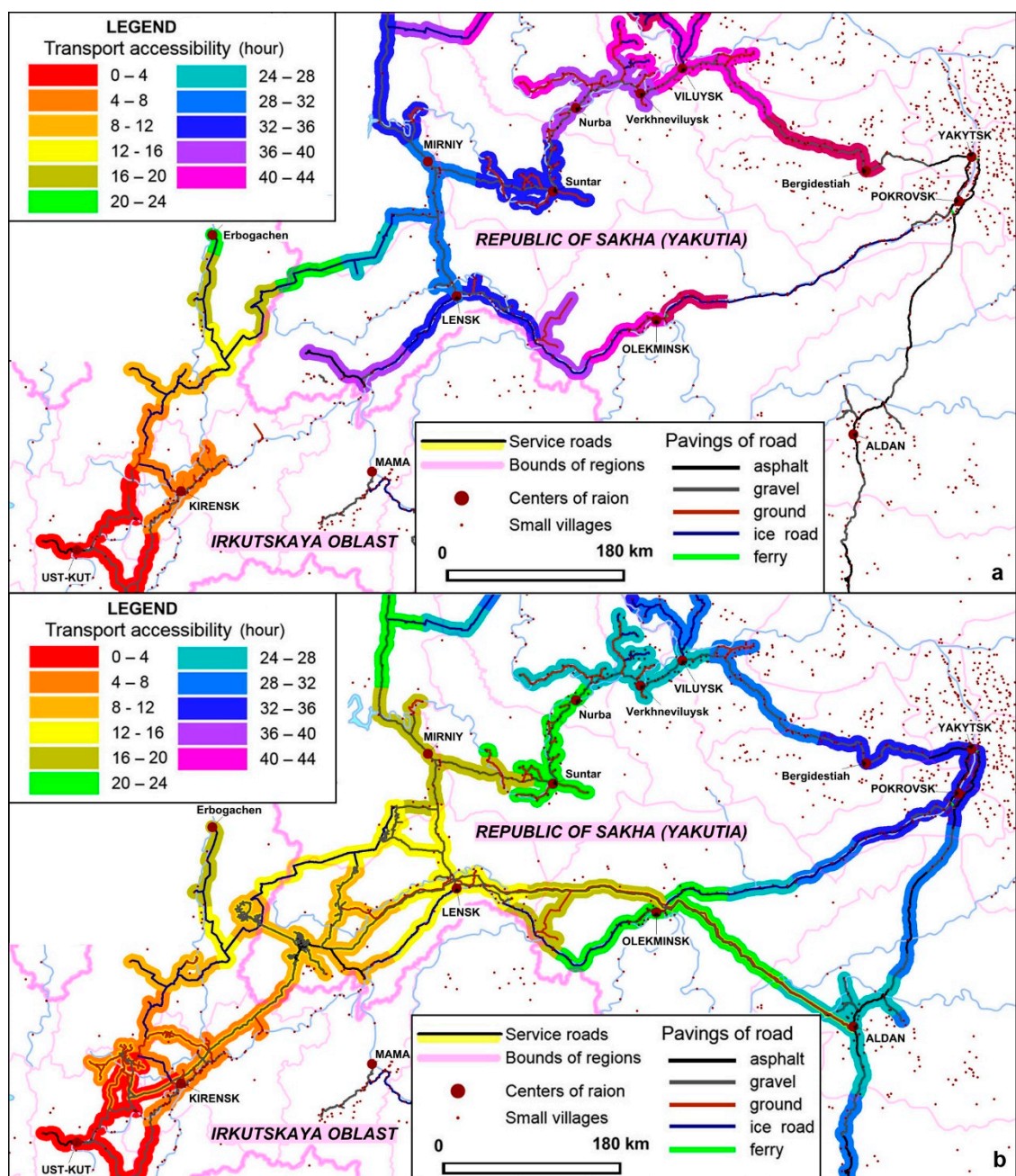

**Figure 2.** Transportation Accessibility of Ust-Kut in winter (**a**) without new service roads, (**b**) with new service roads.

### 3.2. Issues Related to the Informal Road Network Development

Benefit-sharing frameworks that incorporate informal roads, however, may have multiple negative implications for local communities. In general terms, these newly-created issues can be related to access, intrusion and internal community impacts. Implications for access typically encompass the lack or loss of control over and high uncertainty of road use and availability, surveillance and securitization of land use, as well as the necessity to deal with consequences detrimental to the environment and community wellbeing. Intrusion is represented by the onset of newcomer inflows, including poachers, increased levels of disturbance by vehicular traffic, and penetration of other non-local human and non-human intruders (animals and plants). Internal community impacts may be related to the depletion of subsistence resources, uneven mobilities, growing social differentiation, and the loss of land-based

skills and attachment to the land. These problems cut across different benefit-sharing mechanisms, although some are specific for particular arrangements.

Intrusion and rising accessibility to outsiders has been often cited by local residents as a direct consequence of informal road propagation. Heavy machinery used by exploration workers damages local informal roads that already lack maintenance. The workers involved in geological exploration and forestry activities are often blamed for traveling not only for work, but also for fishing and poaching. The local hunters note trash and littering (jars, bottles, plastic bags), oil and fuel from oil drilling machinery licked into the rivers, and even abandoned off-road vehicles [73]. Sometimes the wood left after the forest clearings remains piled on the former hunting trails and even in the river where it was used for river crossing in wintertime. In order to move around, hunters have to clear those debris [74].

Availability of informal roads increased competition for subsistence harvest by opening access to numerous fishermen, who visit the territory not only from the neighboring settlements, but also from distant cities such as Irkutsk. The lakes Agdzheni and Kutukan and the Khanda River are rich with pike, perch and dace [59]. According to local hunters, in summer, there are also occasional fishermen, who travel there, and the number of cars can reach 20 daily. Usually, they have good expensive cars able to traverse low-quality roads. In summer, the outsiders drive to a landing point on the Khanda River 80 km downstream from the village and use boats to move upstream for fishing [72]. While fishing, they often occupy local hunters' huts, stealing and causing damage to the property. Eventually, it led to the practice not to leave anything in the hut. In order to keep non-local fishermen away, one resident had to put barbs around his hunting hut only to find it later burnt down. Another hunter blamed fishermen for accidentally burning his hut. Therefore, hostile relations are formed between 'guest' fishermen and local hunters: "Sometimes you come, your winter hut is full. You start to swear and kick them out" [74].

Increased accessibility of the village also brings a larger number of middlemen who arrive to buy products of traditional activities or engage in harvesting directly. One particular issue is the increased pressure on the pine nuts resources. With the construction of roads and growing volume of traffic, locals typically set up camps and sell pine nuts gathered in the proximity of the roadways. The pine nut harvesting ranges from 50 to 400 kilograms per family and in the situation of high unemployment, pine nut gathering forms a significant share of the family cash income [59]. Increased local harvesting is coupled by the influx of outside gatherers who use the roads to access the taiga. In addition, hunters suspect forest companies and forest agencies in intentionally underestimating the share of Siberian pine nut trees to circumvent the logging restrictions in pine nut tree reach areas because these trees are more expensive in China [72].

A major impact on the ecosystem directly associated with informal roads construction is forest fragmentation. Often the forest is intersected by geological profiles every 150–300 meters forming a tight grid of pathways (Figure 1). In contrast to most official roads, these informal roads are not elevated and often have remaining vegetation. During the dry season travel by heavy machinery by these informal roads can spark forest fires. In addition, easy access to vegetation attracts animals and birds. Almost every interviewee told stories about encounters with bears. Residents often see red deer, and deer crossing the roads and capercaillie and black grouse flying over the roads. As in many other places, these animals and birds are killed on roads under cars or by occasional poaching despite the regulation that only members of indigenous communities and those who have hunting licenses are allowed to hunt in the area. Fishermen are often allowed to carry guns for self-defense since there are occasions when people are attacked by bears. Only the Irkutsk Oil Company has internal regulations prohibiting to carry guns and keep dogs in the area of oil exploration [75].

Informal roads also impact animal migration routes and patterns, and thus affect hunting practices and availability of country food. The impact varies among different species. In particular, hunters note sable returns in a couple of years after the new constructions. There are accounts of moose and red deer moving further to the north using roads. Hunters don't expect moose coming back being substituted

by elk that affect local diet: neither Evenki nor Russian old settlers are accustomed to consuming elk meat [76].

Adapting to the development of informal road networks without means to pay for maintenance and no easy access to repair stations, obshchina members have to be able to repair the cars by themselves. They buy vehicles that can be easily repaired and/or do not require frequent spare parts replacement. Moreover, using gas condensate (provided by oil companies as part of the benefit-sharing agreements) which does not have high quality and damages the vehicles' engines, hunters face the need to repair snowmobiles more often. Vehicles maintenance and travel by informal roads require specific skills: a person should be strong enough to handle the vehicles and make needed adjustments during the travel. Eventually, it is only able-bodied men who travel by those roads, while others (women, children, elders, disabled people) stay in the village.

*3.3. Implications of Specific Benefit-Sharing Arrangements*

Issues associated with the informal road use described in the previous section are common for various types of benefit-sharing arrangements. However, certain benefit-sharing mechanisms exacerbate or generate additional, specific problems.

The negotiated benefits may generate a range of negative effects of road network development. In particular, local communities often have to fully or partially relinquish their transit rights over land and accept disturbance and damage caused by roads. Some hunters have already lost their hunting grounds due to dense extractive infrastructural development. In addition, local communities do not have enough negotiation experience and rarely have access to information and professionals who could assist with formulating and promoting their interests. Meanwhile, the interests of extractive companies are well formulated with the emphasis on benefits the communities will receive. After the agreements are signed and companies received the official community's approval, there is little room for the locals to change the terms. Since the negative effects are usually uncertain and companies' responsibilities for damage are omitted or mentioned vaguely in the agreements, communities don't have access to enforcement and remedy.

Among the issues the local communities face with semi-formal arrangements is the lack of control and high uncertainty on what exactly, how and to whom an extractive company will grant access privileges. The residents make requests that can be described rather as a plea than a demand (cf. [2]). These relationships solidify uneven power relationships between companies and communities and propagate dependency and paternalism. Since semi-formal arrangements typically lack an officially documented proof of access or guarantees to deliver goods and services, local community members face the danger of fines and prosecution for potential 'violations,' while being deprived of legal remedy. In addition, with increased control and monitoring over the users utilizing private roads, the indigenous and local residents become subject to surveillance and violation of privacy. Such securitization of space and scrutinization of mobility disrupt the traditional lifestyle exercised by many Indigenous peoples in the region.

## 4. Discussion

This paper has applied benefit-sharing mechanisms classification framework developed by Petrov and Tysiachniuk [2] to informal roads. The negotiated, semi-formal and trickle-down benefits were identified and illustrated by case studies. Within the negotiated benefit-sharing, informal roads can be linked to compensation payments, special access rights and privileges as a part of benefit-sharing arrangements. As pointed out by Bennett [12], indigenous people can and sometimes do have significant interest in developing transport infrastructure. Their opinions and interests in infrastructural development should be part of negotiations over the development of extractive industries. Moreover, since these roads lay over and often disrupt traditional activities, local communities should have equal rights for their planning and use as a component of benefit-sharing.

The ability to trace the origins of the roads in the remote region in this research is different from the studies of infrastructural violence in the cities where researchers struggle to locate blame and responsibility [13]. In our case, the violence is less abstract and responsible parties are better known. Moreover, often members of local communities are interested in and benefit from infrastructural development. While Saxinger, Krasnoshtanova and Illmeier in their study [16] found disappointment by lack of new road construction and maintenance in another remote community affected by oil extraction and described it in terms of infrastructural violence, communities in our study area did not have permanent access to transportation infrastructure in the first place. Therefore, even the slightest improvements in accessibility are described as beneficial.

Most of the benefits of informal road development are semi-formal and trickle-down. Community members benefit from their own increased mobility as it was already described elsewhere [14,15]. They gain more access to remote parts of the hunting and fishing grounds, become more connected to nearby communities, experience the influx of previously unavailable goods and services, upgrade their fleet of on- and off-road vehicles.

The precarious nature of informal roads makes it difficult to identify their overall impact on local communities. In cases we analyzed, both procedural and distributional equity [77] in respect to roads and benefits is low and power rests with the companies. Little or no legal or other official remedy exists for communities due to the informal and non-public nature of the roads. At the same time, although the informality of road networks gives the power to companies, it may provide the locals with a leverage to 'outsmart' the companies and use roads in adaptive ways. Generation-long knowledge of the land and ability to adapt to changes help the locals to take advantage of the road networks. They enhance traditional hunting and gathering activities using new infrastructure based on formalized or informal relations with companies.

The informal character of developed infrastructure to some extent is beneficial for local communities due to relatively low cost (requirements for maintenance are either non-existent or lower than for regular public roads), flexibility (the ability to change directions and location) of these road networks and lack of state control. With the construction of company roads, transportation by off-road vehicles becomes a new norm for local communities. Since the hunt had been established before and the hunting infrastructure was developed already, the use of new informal roads is not a need, but adaptation to new conditions.

However, the negative impacts likely outweigh those benefits by giving extractive companies incomparably higher benefits and, most problematically, control over where, whom and how to allow access to 'their' infrastructure. With the construction of company roads, transportation by off-road vehicles becomes a new norm for local communities. Increased mobility and intrusion of other actors has brought conflicts, competition for subsistence resources, intrusion of alien species, etc. Those hunters, who do not have access to vehicles in general and especially to the ones of better quality that recreational fishermen have, may feel deprived. Inequalities in access to employment and related financial wealth exacerbate those tensions.

In general, it is already well documented that traditional activities have been conflicting with infrastructural development: newcomers are blamed for game poaching and overexploitation of local biological resources [24–26]. However, the questions of protection of traditional hunting and fishing routes remain complicated by the fact that not only indigenous people, but also other residents are impoverished and rely on subsistence activity. Meanwhile, there is an increasing share of recreational hunters and fishers, who may also originate from local and indigenous communities, but have other sources of income.

Traditionally, indigenous people were hunting and moving around and left behind traces, trails, paths and tracks [78]. For Evenki and old settlers, hunting activities have been part of the traditional way of life and the hunting infrastructure has been developed already. The new transportation infrastructure generated by extractive industries and used in informal ways has become an important factor (and a driver) of their encroachment upon the areas where hunters previously enjoyed with the significant

autonomy of the ways of life. The uncertainties associated with road use rights and responsibilities create power structures that are disadvantageous for local residents and reinforce dependency and paternalism. This is coupled with another unanticipated effect of industrial development: the objectification of nature within the extractivist logic [79]. Encountering outsiders' perception and treatment of animals, plants, and underground as resources, Evenki loose traditions of more intimate reciprocal relationships inherent to their spiritual worldviews and beliefs [59].

## 5. Conclusions

In the article, we examined how streamlined (mandated), negotiated, semi-formal and 'trickle-down' mechanisms of benefit-sharing are materialized in the forms of automobile road network development. This study argues that informal roads constitute an important part of benefit-sharing arrangements, whether intentionally or not. In many areas of intensive extractive activity, they become a prevalent component of natural and social landscapes and exert substantial impacts on social-ecological systems thus require adaptation by local communities. The community-relevant implications of informal roads go far beyond immediate impacts on the surrounding environment, but deeply affect subsistence activities, mobility, food security, personal safety and even consumer preferences of the Indigenous residents. Yet, informal roads are rarely considered as a part of the benefit negotiation process with extractive companies in Siberia. More so, the results of the study demonstrate, most effects of informal road development in remote areas are neither predicted nor monitored.

The lacking role of informal road infrastructure in benefit-sharing frameworks and negotiations determines the high uncertainty of their status and use by different local community members. While community leaders and government officials have more emphasis on benefits, including compensation payments, special access rights and privileges, ordinary hunters express more accounts of environmental degradation, access of recreational hunters and fishermen brought by extractive industrial development. As described in this paper, most arrangements, if any, around the informal roads are made based on semi-formal mechanisms of benefit-sharing (e.g., 'plea- and-take') that are highly precarious, often degrading to the locals and leading to an increased dependency of the Indigenous residents on company's paternalism [8]. However, the informality of these road networks sometimes gives some Indigenous and local residents an opportunity to 'outsmart' the companies and use roads to adapt to the changes. We found evidence of members of local communities being able to adapt to the new roads, adjust their practices and lifestyles (e.g., use of vehicles), as well as establish informal relationships around the new networks.

Informal roads in extractive regions is a widely represented, but poorly studied phenomenon. This is notwithstanding the critical, sometimes transformational role of informal road infrastructure with respect to local communities and the environment. This study only examined three communities to elucidate the nature and outline potential avenues for further research on informal roads as a part of benefit-sharing frameworks between extractive companies and Indigenous/local communities. There was no formal (economic) assessment of impacts or benefits. Therefore, there is limited ability to associate them with particular benefit-sharing arrangements. Further research might include studies in other regions with different climatic, natural, economic and governance conditions.

The focus on specific material elements of benefit-sharing, such as the informal roads, and on how individual members of local communities engage with them, allows to better understand directions in which the local environment, members of local communities and their practices of subsistence activities, mobilities and others change under the impact industrial extractive activities. Therefore, more in-depth studies are needed to understand the impacts of specific informal roads on the environment and local communities and changes in human-environment relations introduced by infrastructural development.

**Author Contributions:** Conceptualization and methodology, V.K. and A.N.P.; investigation and data analysis, V.B., N.K., V.K. and A.N.P.; writing—original draft preparation, N.K., V.K. and A.N.P.; writing—review and editing, A.N.P.; mapping and visualization, V.B.; project administration and funding acquisition, V.K. All authors have read and agreed to the published version of the manuscript.

**Funding:** This article is the resulting effort of works on the research project "Informal Roads: The Impact of Unofficial Transportation Routes on Remote Arctic Communities" supported by the National Science Foundation (#1748092), and the research project 'Configurations of "remoteness" (CoRe)–Entanglements of Humans and Transportation Infrastructure in the Baikal-Amur Mainline (BAM) Region' (FWF) [P 27625] at the University of Vienna, Austria. This study also benefited from RCN Arctic-FROST workshops (National Science Foundation #1338850).

**Acknowledgments:** The research team is grateful to the informants who shared their time, stories and expertise. This article is partly based on research conducted by Gertraud Illmeier and Natalia Krasnoshtanova in 2016 and 2018 in Tokma.

**Conflicts of Interest:** The authors declare no conflict of interest.

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
