# Peer review of "Mobilizing Benefit-Sharing Through Transportation Infrastructure: Informal Roads, Extractive Industries and Benefit-Sharing in the Irkutsk Oil and Gas Region, Russia"

_resources, doi:10.3390/resources9030021_

Round 1
Reviewer 1 Report
Spell out: CSR and SLO
Why this is Italicized? mechanisms of benefit implementation
Can you frame this in terms of importance in countries other than Russia? This would make the article more interesting for an international audience.
I am guessing that transportation infrastructure refers to roads and bridges. Transportation infrastructure could also be trains or bus stops. Can you ore specific or define at the begging what is meant by transportation infrastructure?
Tabl2 2, hard to read because of how the cells are formatted (justified?)
Can you say more about the interviews, which kids of questions you asked, how the interviews were analyzed, did you use grounded theory for the analysis?
There is a lot of imbalance between the 3 geographies and the number of interviews. 3 interviews for Vershina Khandy show little evidence. Why in his community the authors did not interview Company Employees (CE) or Gov. officials? I would be more convinced that these 16 interviews were juicy if there were more quotes or more background of what it was done and why. The authors should also discuss the limitations.
Figure 2, can you fix this image, it is blurry.
The conclusion is weak, make sure to go back to the interviews and what they show, mentioned previous studies and what is the contribution of this piece.
Author Response
Thank you very much for the helpful review.
Spell out: CSR and SLO - fixed
Why this is Italicized? mechanisms of benefit implementation - removed formatting
Can you frame this in terms of importance in countries other than Russia? This would make the article more interesting for an international audience. - the introduction we discuss the problems of benefit sharing and infrastructural development common for many countries.
I am guessing that transportation infrastructure refers to roads and bridges. Transportation infrastructure could also be trains or bus stops. Can you ore specific or define at the begging what is meant by transportation infrastructure? - added text on the page 2
Tabl2 2, hard to read because of how the cells are formatted (justified?) - removed formatting
Can you say more about the interviews, which kids of questions you asked, how the interviews were analyzed, did you use grounded theory for the analysis? - added
There is a lot of imbalance between the 3 geographies and the number of interviews. 3 interviews for Vershina Khandy show little evidence. Why in his community the authors did not interview Company Employees (CE) or Gov. officials? I would be more convinced that these 16 interviews were juicy if there were more quotes or more background of what it was done and why. The authors should also discuss the limitations. - added explanation
Figure 2, can you fix this image, it is blurry. - fixed
The conclusion is weak, make sure to go back to the interviews and what they show, mentioned previous studies and what is the contribution of this piece. - have added several sentences in conclusions to elaborate our main points.
Reviewer 2 Report
Benefit sharing in extractive industries is most important way how the right to permanent sovereignty over natural resources can be implemented by local and indigenous people within their territories. Up-to-now, it is mostly omitted by the state authorities and administrations that not the state but peoples are subjects of the right to sovereignty over natural resources, as defined by the UN resolution 1803 (XVII) of 1962. From this viewpoint, the manuscript submitted solves rather marginal questions of costs and benefits of informal roads constructed by extractive corporations, as a potential part of benefit sharing for local inhabitants. In spite of that, article brings, on the basis of three community case studies from Irkutsk oil and gas region, interesting information about how deeply informal roads affect and degrade subsistence activities of local communities.
Some abbreviations remain unexplained
2, line 6: there are have been 3, l. 34: neither exist officially, not mappedAuthor Response
Thank you for your helpful feedback.
Benefit sharing in extractive industries is most important way how the right to permanent sovereignty over natural resources can be implemented by local and indigenous people within their territories. Up-to-now, it is mostly omitted by the state authorities and administrations that not the state but peoples are subjects of the right to sovereignty over natural resources, as defined by the UN resolution 1803 (XVII) of 1962. From this viewpoint, the manuscript submitted solves rather marginal questions of costs and benefits of informal roads constructed by extractive corporations, as a potential part of benefit sharing for local inhabitants. In spite of that, article brings, on the basis of three community case studies from Irkutsk oil and gas region, interesting information about how deeply informal roads affect and degrade subsistence activities of local communities. - we have added text to explain why informal roads are not only an interesting phenomenon, but also an important element of benefit sharing arrangements that is often overlooked.
Some abbreviations remain unexplained - spelled out
2, line 6: there are have been 3, l. 34: neither exist officially, not mapped - did not understand the question / comment
Round 2
Reviewer 1 Report
There are two sentences that need to be cited to avoid plagiarism (in bold the same text as other online): Benefit-sharing is formally defined as the distribution of monetary and non-monetary benefits generated through the resource extraction activity [1,2]. From: https://www.mdpi.com/2079-9276/8/3/155/htm?
northern parts of the Irkutsk region and the Republic of Sakha (Yakutia). It is known as the ‘northern supply’. The main sources of income of the local population are the public sector, transportation, forest industry and the service sector. From: https://www.tandfonline.com/doi/full/10.1080/1088937X.2018.1564395
Author Response
The following change were made in the document:
As it was suggested, we have inserted the reference in the sentence "Benefit-sharing is formally defined as the distribution of monetary and non-monetary benefits generated through the resource extraction activity [1,2]. "
And the sentence "The main sources of income of the local population are the public sector, transportation, forest industry and the service sector" was re-written: "Local residents have been employed in the public sector, transportation, forest industry, and services"
However, the sentence has citation of the source published earlier than the source mentioned by the reviewer. And the following information was published there: "The main sources of income for the majority of local residents are the public sector, transportation, forest industry, and services"